# Flaminio Rota: Fame and Glory of a 16th Century Anatomist without Scientific Publications

**DOI:** 10.3390/ijerph18168772

**Published:** 2021-08-19

**Authors:** Gianfranco Natale, Paola Soldani, Marco Gesi, Emanuele Armocida

**Affiliations:** 1Department of Translational Research and New Technologies in Medicine and Surgery, University of Pisa, 56126 Pisa, Italy; paola.soldani@unipi.it (P.S.); marco.gesi@unipi.it (M.G.); 2Museum of Human Anatomy “Filippo Civinini”, University of Pisa, 56126 Pisa, Italy; 3Center for Rehabilitative Medicine “Sport and Anatomy”, University of Pisa, 56126 Pisa, Italy; 4Department of Medicine and Surgery, University of Parma, 43126 Parma, Italy; emanuele.armocida@studio.unibo.it

**Keywords:** Flaminio Rota, University of Bologna, human anatomy, university teaching, university research, history of medicine

## Abstract

Academic activity is intrinsically composed of two aspects: teaching and research. Since the 20th century, the aphorism “publish or perish” has overwhelmingly established itself in the academic field. Research activity has absorbed more attention from the professors who have neglected teaching activity. In anatomical sciences, research has focused mainly on ultrastructural anatomy and biochemical aspects, far removed from the topics addressed to medical students. Will today’s anatomists be rewarded by their choice? To generate a forecast, we should entrust what history has already taught us. For this analysis, an example was taken, concerning the fate that history reserved for the anatomy teachers of the University of Bologna in the second half of the 16th century. Thanks to Vesalius (1514–1564), experimentation on the human body replaced the old dogmatic knowledge, and didactic innovation was one with research. Some figures were highly praised despite their poor scientific production. The present article focuses on the figure of Flaminio Rota, who was highly esteemed by his colleagues in spite of no significant scientific activity. Reasons for this paradox are examined. Then, history also whispers to us: publish, but without perishing in the oblivion of students.

## 1. Introduction

After the invention of the printing press by Johannes Gutenberg in the 15th century, the circulation of information increased exponentially, leading to cultural and social changes. In addition to this, Galileo Galilei (1564–1642), considered the father of the scientific method, introduced procedural rules in carrying out experiments, and underlined the importance of disseminating the results of the research. Thus, the modern scientist is a figure who conducts experimental research and publishes their results to improve knowledge in a field of interest.

Since its medieval origin, the university represented the highest level of cultural institution and the seat of research and teaching. For this reason, the fame of university professors was mainly associated with important discoveries, published in Latin (and afterwards, in national languages) to have the greatest dissemination. Throughout the Middle Ages until the late 16th and early 17th century, education in western Europe was organized by the Catholic Church, with Latin as the language of learning. Therefore, it is not surprising that the language of communication in science was Latin. The technology of the printing press set in motion far-reaching developments; the number of books in the vernacular (Dutch, English, French, German, Italian, and Spanish) increased in comparison with the number of Latin publications, thereby eroding Latin hegemony. In general, literacy in early modern society increased, which meant that authors could presume a larger audience to transmit the practice of science. Finally, over the centuries, authors have returned to writing in the most widespread language in the scientific community to gain visibility; for example, German in the 19th century and English today [1].

In science and technology, the eponym is the clear expression of this situation, when a discovery or an innovation can be named after the scientist who has demonstrated their determinant role. The history of science includes several cases of diatribes and disputes for the right attribution of a discovery. Indeed, according to Stigler’s law of eponymy, the priority of a discovery is often not easy [2,3].

When working on one’s own or without publishing, the risk of losing one’s merits is very high. The most famous instance of this is in the case of Leonardo da Vinci. Although well known for his ingenuity, Leonardo published nothing and all his research survived only in dispersed and confused manuscripts; the most part of his discoveries and observations was recognized posthumously.

At present, *publish or perish* is a ruthless tongue-twister aphorism describing the pressing need to publish scientific works in order to gain priority over others, to obtain financial support, and to succeed in academic careers. Although this aphorism was coined about a century ago, it has only gained relevance [4].

However, in the past, some figures have been highly praised despite their poor scientific production. In particular, the present original article focuses on the figure of Flaminio Rota, a surgeon and professor of anatomy in Bologna during the 16th century. Rota was highly esteemed by his colleagues and several honors were bestowed upon him. In spite of this, no significant scientific activity attributable to Rota was found. Therefore, Rota did not publish relevant original research and did not perish. Possible reasons for this paradox are examined.

## 2. Rota’s Life and Career

Information about the early life of Flaminio Rota is lacking. His date of birth is unknown, although, in the Archivio di Stato of Bologna, a document states that he was baptized on May 6th, 1555, in Saint Peter’s cathedral of Bologna [5].

His father, Giovanni Francesco Rota (1520–1558), graduated in Philosophy and Medicine from the University of Bologna in 1546–1547, where he became a professor of anatomy and surgery. He joined the papal army as surgeon and participated in the siege of Parma and Mirandola in 1551. His activity and works were particularly considered. In 1553, he published *De introducendis Graecorum Medicaminibus Liber* [6] and, in 1555, he published his most important work, *De bellicorum tormentorum vulnerum natura et curatione liber*. The latter publication was derived from his experience as military surgeon, and a second edition was printed in Venice in 1556, including the treatise *De vulneribus sclopetorum*, which also appears in subsequent editions. Indeed, the main interest of Giovanni Francesco was the treatment of wounds provoked by firearms. This was a new field of research, and he was a pioneer in recognizing the burning effect of firearms. He also described the effects of contusions and bomb explosions. A wooden bust of Giovanni Francesco Rota was placed in his memory inside the famous anatomical theatre of the Archiginnasio of Bologna (Figure 1). In that time, the famous French surgeon Ambroise Paré (1510–1590) was also a pioneer in surgical techniques, battlefield medicine, and the treatment of wounds.

Following the career of his father, Flaminio also graduated in Philosophy and Medicine at the University of Bologna on March 8th, 1577. Thanks to his family origins, he managed to make his way into all the most important medical institutions in the city. From 1579, he was lecturer, and from 1589–1590, he also taught anatomy, until his death in 1611 [7,8,9]. His academic experience coincided with important cultural changes in the approach to anatomical studies. From the Renaissance and into the 16th century, Galenism and Scholasticism were increasingly out of favor, and open-minded observation and experimentation on the human body started to replace the old anatomical knowledge based on dogmatic doctrines [10].

The rebirth of the anatomical science was started in Bologna in 1315 by Mondino de’ Liuzzi (1275–1326), the “restorer of anatomy”, with the rediscovery of public dissection. After the glorious Alexandrian season of Herophilus and Erasistratus (3rd century BC), the systematic dissections of human cadavers ceased, and poor progress was made in the study of human anatomy; physicians could, therefore, only follow the works of eminent figures of the past, such as Aristotle or Galen, without seriously questioning their scientific validity. Mondino’s *Anothomia* was completed around 1316 and, due to the clarity of his text, became the reference book in nearly all European medical schools for the next three centuries. However, Mondino’s teaching method was demonstrative. Experimental anatomy, centered on observation, began thanks to the famous Flemish anatomist Vesalius (Andreas van Wesel, 1514–1564), with the publication of his famous *De Humani corporis fabrica* in 1543. Then, anatomical teaching moved from conservatism to innovation, and the frontispiece iconography of Vesalius’ *fabrica* showed this revolution. Furthermore, the popularity of anatomy was not confined to physicians or medical students but also involved artists, such as Leonardo da Vinci and Michelangelo Buonarroti [11,12]. Hence, it appears evident the important role of different institutions (universities, schools, academies) in teaching and research activities. As far as anatomy is concerned, public or private lessons and demonstrations were available. Scholarship and publications depended on the cultural level and fame of professors and practitioners [13].

When Rota started his academic activity, Giulio Cesare Aranzio (1530–1589) was in charge of anatomical teaching. Aranzio became a significant ambassador of intense experimentation in northern Italy. He was a student of Vesalius and played a revolutionary role in the official acknowledgment of the teaching of anatomy as a distinct subject. The teaching of anatomy at Bologna was mostly based on the performance of public dissections, but it was somewhat irregular and there was no Chair of Anatomy. The Statute of 1405 established that “any doctor, requested by the students, should perform the anatomical dissection”. The provisions of the Statute remained in effect until 1570. In 1570, the officials of the Studium or Senate approved a decree that changed the status of the teaching of anatomy. This decree established that there should be a Chair of Anatomy and an ordinary professor appointed to that chair, separate from the Chair of Surgery. For the first time, another heading—AD anathomiam ordinariam, Julius Caesar Arantius—appeared near the name of Aranzio as AD lecturam chirurgiae. This event was a big step forward in the history of anatomy, although the separation was not absolute until the middle of the 17th century [14].

In Bologna, other important surgeons were Rota’s colleagues at that time, including Gaspare Tagliacozzi (1545–1599), a pioneer in plastic surgery, Girolamo Mercuriale (1530–1606), who renewed the therapy of syphilis, and Giovanni Battista Cortesi (c. 1553–1634), known for his pharmacopeia and anatomical studies on the central nervous system. The renown of the Bologna medical school was based not only on the fame of its teachers, but also on the functionality of the places dedicated to the teaching of medicine and anatomy, in particular the so-called Anatomical Theatre. The construction of a room suited to the teaching of anatomy, where corpses could be dissected for teaching purposes, similar to the ones that had been built in Padua or Pisa [15], was urged by the Bologna Senate from 1595 [16].

Dissections that took place within its walls, and which certainly saw Rota among the first protagonists, combined cultural, philosophical, and medical experiences that have changed uses and customs of society [17,18].

As confirmed by archival documents, Rota started his lectures in 1579–1580 *Ad lecturam chirurgiae*. By 1586 lectures in surgery had a dermatologic approach and were based on Galen’s tripartite work: (1) *De tumoribus praeter naturam*; (2) *De ulceribus*; (3) *De vulneribus* [19]. Then, for seven years (from 1579–1580 until 1585–1586), Rota did not lecture on established arguments, but he did give free surgical lectures. Finally, in 1586–1587, for the first time, he based his lecture on *De tumoribus praeter naturam*; in 1587–1588, on *De ulceribus*; and in 1588–1589, on *De vulneribus*. He followed this order, but in 1589–1590, besides *De tumoribus praeter naturam*, he also based his lecture on *Ad anathomiam* [20].

In 1595, Rota entered the prestigious Collegio di Filosofia e Medicina. The participation to this selected college represented an important status symbol in Bologna and allowed a special contact between political and economic circles. Furthermore, since, in Bologna, a medical corporation was lacking in that time, the college also had the role of handling professional applications [21]. As a representative of this college, in 1606, Rota was signatory, together with Melchiorre Zoppo, of an agreement with the Compagnia degli Speziali Medicinalisti in Bologna. The document was signed by Ercole dal Bono and Orazio Campioni (Compagnia), Giovanni Banoso (representative of the university of pharmacy), and Alessandro di Sangro, vice-legate of Bologna. It regulated the activity and authorization of the chemist for drug preparations and his relationship with physicians and persons deputed to control the right preparation of drugs. Finally, the decree also included *Index rerum medicinalium*, a list of therapeutic compounds [22]. The presence of Rota and Zoppo in the college is confirmed in the first lines of the license to drug prescription included in the *Liber pro recta administratione protomedicatus*, published in 1666 [23].

According to Alidosi [24], Rota died on January 16, 1611, and was buried in Bologna in the church of San Giacomo Maggiore.

## 3. Teaching Activity

Rota was undoubtedly a praised teacher. The Palace of Archiginnasio in Bologna preserves the best evidence of the recognition of his didactic activity [8,25,26]. This palace was built in 1562–1563, thanks to the papal legate in Bologna, cardinal Carlo Borromeo, and vice-legate Pier Donato Cesi, upon the project of the architect Antonio Morandi, known as Terribilia. Under the cultural climate of the Council of Trent, this initiative aimed to provide a unitary seat to university teaching. Today, in the Archiginnasio, visitors can admire an extraordinary collection of coats of arms of the different colleges, as well as marble epigraphs in memory of professors particularly appreciated by students. In this respect, when alive, Rota was praised with six memorial epigraphs. To have an idea of the comparative value of this recognition, it must be considered that the famous anatomist Aranzio deserved eight epigraphs. Rota was appreciated because he radically innovated the anatomical teaching with interactive lessons and open discussions. Furthermore, according to Vesalius’ revolution, he gave importance to anatomical dissection. Thus, in spite of the countless inscriptions that decorated—and, in part, still decorate—the walls of the Archiginnasio, many celebrated the art of such contemporary anatomists as Flaminio Rota or Cortesi, who, in the course of public anatomy lessons, had found rapid solutions to “the very difficult objections raised extemporaneously by distinguished scholars” [26].

Alidosi [24] and Medici [8] reported the following epigraphs found in public schools:
*H. M. M. AE.**Praeclaris. Flaminio Rotae Philosoph. et Medic. Doct. ingeniosis. Ordinariae anatomes perspicaciss. investigator istructorique solertiss. Munere egregie perfuncto electores Syndici ac Philosoph. et Medic. Consiliarii grati animi ergo P.C.C.**Gloria virtuti merces ac fama perennis**Partaque pro meritis proemia iustus honos.**Haec Rota iure tibi delata fatentur alumni**Doctrinae hic testis tempus in omne lapis.**Tu Cui monumenta senis, tu scripta Galeni**Sic aperis sensusque eruis anatomes.**Ut cedat tibi Phyllirides doctusque Machaon**Ut cedat medica quisquis in arte valet.**Anno sal. hum. MDXC. XIV. cal. Martii.**Eidem Flaminio Rotae.**Viro doctissimo solertissimo eloquentissimo, quo anatomen publice administrante quod iam diu quotannis summa cum sui laude descintiumque utilitate docendo disserendo incidendo perfecit ipsam de se loqui naturam naturaeque ministram artem medicam facile existimes auspice Illustr. D. Io. Dom. Spinula Rectore Generali lapis positus.* (Figure 2)*H. M. M. AE.**Clariss. Philosophoac Medico Flaminio Rotae.**Quod in summa die culte de decretiantium onus anatomes explicanda interpretandi negotium susceperit rescindendi laborem non defugerit.**Quod virorum obiecta ex tempore prompte adeo soluit omniaque ad anatomicam inspectionem spectantia problemata ingeniose adeo, ac sapienter enodavi, ut inde non minimum gloriae dignitatisque sibi comparaverit, musis debita iuventus in tanti meriti memoriam exiguum hoc mausolaeum fieri constituit. Annuentibus D. Gentile Melzio Arimin. Priore, ac D. D. Cas. Rosetto Forolivien. RochoIordano Tridentino Ellect. MDXCIX.**Flaminio Rotae Philosopho ac Medico Bonon. qui cum plures annos humani corporis dissectiones cum publice tum privatim diligentissime obeundo, elegantissime omnia enarrando, ex temporalibusque in disputationibus dubia, atque obscura eximie illustrando praestantissimi cuiusque in hoc genere gloriam adaequavit electores, et anat. synd. hoc grati in eum monumentum P.C. sub. Foelicibus auspiciis. Illust. et generosi D. D. Io. Ferrarini Saxolensis Prioris, digniss. Annuentibus Illust. D.D. Nestore Cantuto Mutinensi Dom. Petro Paulo Galliardo Vitelianensi elect et Anot. synd.**An. Sal. M DC II. idib. Martiis.*

In this respect, Rota’s students represent the best confirmation of this appreciation. The medical school of Bologna accepted international students, and then Rota trained physicians all through Europe. Two Italian and two foreign students are mentioned.

Cesare Magati (1579–1647) was one the most famous students of Rota [27,28,29]. He is considered an innovative figure in wound treatment, and, in 1616, he published *De rara medicatione vulnerum* [30]. As student, he learned new techniques during Rota’s lessons and trained in the Hospital of Santa Maria della Consolazione. He has even been compared to Rota for scientific and personal reasons. However innovative Rota might have been, his adherence to Hippocratic medicine was unchanged. In some letters written in 1610 and preserved in the Accademia dei Concordi of Rovigo—one of the several *accademie* founded in that time to protect and disseminate culture—a dispute arose on how Magati treated a cephalic wound of a German officer. In the first letter, Magati called Rota to account about some judgments: “…V.S. Ecc.ma haveva detto, che in detta cura io haveva commesso errore e che il ferito era morto per mia causa, per haver io tralasciato gli opportuni rimedii per la sua salute… [His Excellency had said that in this treatment I had made a mistake and that the wounded man had died for my cause, for having neglected the appropriate remedies for his health]”. Rota replied that he was informed that the unknown physician treated according to the “roman method” and that the best method is the hippocratic one: “… A’ che io risposi, che questo metodo non mi piaceva, non l’Havere per sicuro; ne meno era riuscito in Bologna a qualche d’uno ch’haveva voluto tentarlo … chi non medicherà le ferite alla testa con Hippocrate, difficilmente guarirà … [I replied that I did not like this method, as it was not safe; others in Bologna had also failed … those who do not treat head wounds with Hippocrates will hardly recover]” [31].

Besides being a student, Giovanni Capponi (1586–1628) was also a guest in Rota’s house, which was usual in that time [32]. He mainly dedicated himself to political sciences, but thanks to his medical studies in 1620, he published *Lettura di Parnaso e Discorsi Accademici* [33]. In this work, he interpreted the first of Hippocrates’ aphorisms in a political perspective, but the concepts were attributed to Rota. Capponi probably chose the figure of his master to diffuse his ideas because he was more authoritative and influential. In particular, Capponi imagined Rota during summertime when releasing private lessons about the first aphorism to “Studenti di Medicina della difficilissima Ragion di Stato” [34].

Two foreign students were Flaminius Gasto (1571–1618), who graduated in 1597 [35], and Pieter Rapaert II (1579–1637), a famous physician in Bruges who graduated on September 16th 1605 [36].

## 4. Surgical Activity

Other than an appreciated teacher, Rota was also a skillful surgeon. From 1576 to 1580, he was student assistant (so called astante) at the Hospital of Santa Maria della Morte in Bologna. The student assistant was subordinate to the physician and the surgeon, was elected by the same hospital administrative board, and had to be “a student of modest means, greatly promising, and likely to become a very good physician, diligent and full of charity”. Formally, the assistant was elected by a vote among the members of the officials of the hospital and the members of the brotherhood, so presumably this election was exposed to the same systems of micro-power that were in place for the election of the physician and the surgeon. In that period, the list of student assistants also included Giulio Cesare Gessi (1570–1576) and the above mentioned Tagliacozzi and Cortesi. All of them were lecturers in surgery and anatomy. The student assistant had in his turn a number of people who were his subordinates: the nurses, over whom he could exercise his power within a precise hierarchical organization. The student assistant was also caught in the process of rapid transformation and definition of the medical staff, which started in the second half of the 16th century. The general direction of the transformation was towards increasingly specific roles: the student assistant became more and more specialized in diagnostic and observational tasks, and in taking care of the surgical tasks of first aid [19,37].

In the Hospital of Santa Maria della Morte, contagious or incurable patients were not admitted. The following diseases were considered contagious: leprosy, syphilis, scabies, plague, and similar. The following diseases were considered incurable: inveterate cancer, sciatica, hepatic flux, pain in the junctures, dropsy, gout, and similar. The Hospital of San Giobbe degli Incurabili (previously known as the Hospital of San Guerino dei Guerini, then of Santa Maria dei Guerini, of San Lorenzo dei Guerini, and Santa Maria delle Laudie) at first was conceived to cure and assist pilgrims. Because of the diffusion of syphilis in Bologna in the late 15th century, it was then designed to receive patients affected by this new infection, becoming one of the first hospitals specialized in venereal diseases. It was active until the 18th century, when it was finally incorporated into the Hospital of Sant’Orsola. From 1578 until his death, Rota was active as surgeon in the Hospital of San Giobbe. According to the statute of the hospital, it was established in the presence of a surgeon and a physician, the latter being more important. The surgeon was responsible for the treatment of wounds. The application of ointments and other drugs was under the control of the physician. At that time, the physician was Enea Vizzani (also known as Aeneas Vigianus, Vizanus, or Vizanius). When he died, the role of the physician was not replaced, and Rota was active alone. After Rota, the figure of the physician was not requested anymore [38].

In 1585, Rota was named “supernumerary surgeon” at the hospital of Santa Maria della Vita [5,19].

## 5. Scientific Activity

Rota was interested and involved in different anatomical arguments, but he did not publish any scientific research. Nevertheless, indirect information about his activity can be found in other works, where some of his studies were admirably mentioned by his contemporary colleagues.

One of these examples is represented by the famous naturalist, botanist, and entomologist Ulisse Aldrovandi (1522–1605). One of his last publications was *Ornithologiae hoc est de avibus historiae*, in three volumes, in which he reported notes on anatomists who also carried out studies dealing with animal and comparative anatomy. In particular, Rota and Varolius were praised by Aldrovandi [39] (volume I, book XII, chapter XVIII, p. 799) for their studies on *Bombycilla garrulous* (Bohemian chatterer, now Bohemian waxwing): *Secuerunt hanc avem clarissimi viri, Constantius Varola, & Flamminius Rota, ambo in celeberrimo nostro gymnasio Anatomiae professors publici, & viri celeberrimi, mihique amicissimi*. These notes were also mentioned by Craigie [40] (p. 361): “Both these anatomists supplied, by their researches, much anatomical information to Ulysses Aldrovandus”. Similarly, Aldrovandi [41] (volume II, book XIII, p. 92) mentioned Aranzio for his research on the morphology of *Otis tarda*: *Secuit olim mihi hanc avem clarissimus Iulius Caesar Aurantius, quo in secandis corporibus nemo unquam peritior, nemo in explicandis disertior extitit*.

Vincenzo Alsario dalla Croce (1576 ca–1632) published *De epilepsia, seu comitiali morbo* [42] (p. 91), in which he mentioned the anatomical studies performed by Rota, praised as “percelebris Anatomiae professor” (very famous professor of anatomy).

Other than a pioneer in forensic medicine in Italy, Giovanni Battista Codronchi (1547–1628) was also a skillful anatomist. In a work dedicated to the luxation of the xiphoid process of the sternum, *De morbo novo, prolapsu scilicet mucronatae cartilaginis dicto* [43] (p.53), he reported the opinion of the “excellentissimus medicus” (very excellent physician) Rota, that the process can undergo only an improper luxation downwards.

Again, the German physician Daniel Sennert (1572–1637), in his work *Operum tomus II* [44] (p. 447), considered Rota “anatomicus apprimè peritus” (very expert anatomist), when discussing the cartilage luxation.

In his work *Consultationes, in quibus universa praxis medica exacte pertractatur* [45] (pp. 286–293), the famous clinician Giovanni Zecchi (1533–1601) included a case (n. LVIII) transmitted by Rota referring to the Bolognese politician Giacomo Maria Campanacci suffering from bladder stones: “De colli vescica esectione ad tollendos dolores lythyasi laborantes in reddenda urina crudeliter excruciantes, pro Perillustri D. Iacobomaria Campanacio illustrissimi Senatus Boniensis a secretis maiore cancellario ex eccellentissimum D. Flaminium Rotam Medicum Celeberrimum”. After an accurate examination of the case-history, Zecchi decided to remove calculi by lithotomy.

## 6. Discussion and Conclusions

The main flaw in Rota’s life is the lack of scientific publications. Medici [8] evidenced that some historians neglected Rota’s memory just because of this fault. Nevertheless, this was not enough to completely obscure his career. In fact, in *Memoriae medicorum nostri seculi clarissimorum renovatae decas prima* (1676, p. 56), Henning Witte (1634–1696) even mentioned Rota as a very famous Italian medical figure, together with Galileo Galilei and Santorio Santorio [46].

In past times, the modern aphorism *publish or perish* was not so compulsory and determinant. In the late 16th century, only the lower-status learned surgeons were moved and pressed to publish on their practical activities as surgeons and anatomists to gain advancement and to establish their names. The University of Bologna offers an example of such a situation. In that time, five surgeons had different backgrounds: Aranzi and Tagliacozzi came from the artisanal class and Cortesi was born to a poor family; on the contrary, Rota and Angelo Michele Sacchi (1538–1611) were part of medical families of collegiate physicians. Not surprisingly, both Rota and Sacchi were the only ones not to publish. Then, “Rota built his career upon his ability as a teacher, both in private and in public, and by exploiting the social prestige he inherited from his family tradition” [19].

Accordingly, Rota had no time and no need to publish his studies. Rather, to save his scientific observations, he communicated them to some famous colleagues and he shone with reflected light. It must also be considered that the anatomical research of Rota was too heterogeneous and not sufficiently organized to be enough for a systematic publication, and it can be hypothesized that Rota was aware of this. Today, the most part of scientific research consists of short articles published on dedicated journals. In Rota’s time, this opportunity did not exist. Alternatively, epistolary communications with colleagues and letters included in books were a possible way to advertise a discovery. However, Rota did not adopt that possibility, either. This means that he fully refused to publish or to compete on scientific subjects.

Until approximately two centuries ago, the book was the principal vehicle to report scientific observations and discoveries. In 1665, *Philosophical Transactions*, the world’s first and longest-running scientific journal, was launched by Henry Oldenburg. Since then, scientific peer-reviewed journals have become the most common way to deliver information within the scientific community, and the scientific article has become the basic unit to measure the quality of research.

In recent years, the entry of market analysis into the academic publication system has determined the emergence of research incentives and agency accountability. In particular, the business-like model of the *publish or perish* rule intersects today with commercial strategies that alter the choices of researchers and the objectives of the research itself, leading to negative phenomena, such as predatory journals and publishers.

Several approaches and parameters, including literature-driven methods, the golden standard Impact Factor conceived by Eugene Garfield (Journal Citation Report), Immediacy Index (ISI-Thomson), H-index, the Matthew effect, university and journal ranking, open access, and bibliometric methods, have been introduced in an attempt to measure the quality of research and to ameliorate modern scholarly publishing. Nevertheless, at the same time, they have also led to a disruptive innovation.

In addition to this, in recent years, universities have been firmly invited to shift from mainly teaching and performing research in order to take responsibility for the so-called third mission, in terms of multidisciplinary contribution to society. This new mission involves the engagement of non-academic stakeholders and includes several tasks to create a more favorable environment: transferring knowledge and technologies to industry (joint collaboration with industry, academic spin-off creation, and patenting); triple helix model partnerships; knowledge transfer and entrepreneurship education; socio-economic development and welfare; employment; and quality of life [47].

Considering teaching activity, which was so important for Rota, at present, a renewed attention is dedicated to modern learning methods. For example, medical informatics education is growing for the acquisition of digital competencies with interdisciplinary and collaborative teaching teams, as well as biomedical modelling and simulation. These projects must be integrated into training curricula to prepare physicians for modern healthcare, and are addressed to both undergraduate [48] and postgraduate medical students [49]. In this frame, the teacher plays an important role in coordinating and managing information with new techniques, leading to a greater accountability in education and an increased professionalism by creating higher standards of practice.

Discovering the biography of Flaminio Rota, this study highlighted how teaching is a fundamental activity and characterized the profession of university teacher. Not only that, but the teaching of knowledge is the most sought after, appreciated, and warmly remembered quality by students. Finally, history whispers to us: publish, but without perishing in the oblivion of students.

## Figures and Tables

**Figure 1 ijerph-18-08772-f001:**
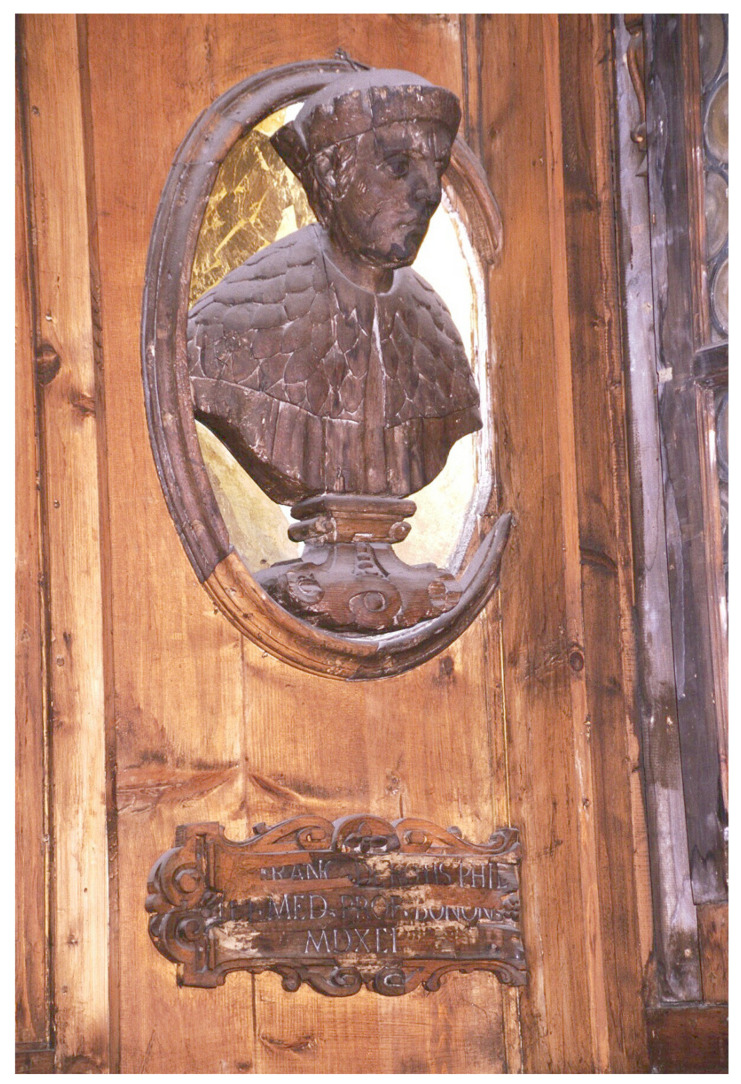
Wooden bust of Giovanni Francesco Rota in the anatomical theatre of the Archiginnasio Palace of Bologna.

**Figure 2 ijerph-18-08772-f002:**
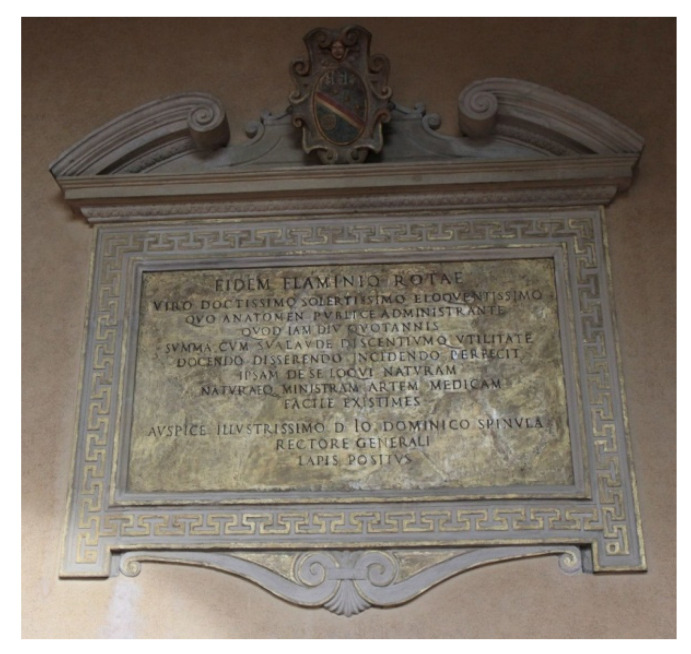
Flaminio Rota’s memorial tablet in the east side of the upper loggia of the Archiginnasio Palace in Bologna (1590). Source: http://himetop.wikidot.com/ (accessed on 25 June 2021).

## Data Availability

Not applicable.

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
