# Peer review of "Flaminio Rota: Fame and Glory of a 16th Century Anatomist without Scientific Publications"

_ijerph, 2021, doi:10.3390/ijerph18168772_

Round 1

Reviewer 1 Report

The text is very interesting and well written. It manages to give a fairly exhaustive picture of the period in which Flaminio Rota lives and of the major scientific figures of the time.

The only note I can make is to have inserted redundant information about the 'anatomical theater' and the 'figure of the assistant', which interrupt the discussion of the topic and do not make it easy for the reader to read.

Finally, it would be useful to include in the bibliography some references relating to the question today much debated of the methods of teaching medicine. This would give a scientific basis to the statements, which in my opinion are quite correct, relating to the issue of teaching/research/publications.

Author Response

Referee 1

The text is very interesting and well written. It manages to give a fairly exhaustive picture of the period in which Flaminio Rota lives and of the major scientific figures of the time.

The authors would like to thank the reviewer for the words of appreciation regarding the manuscript. Changes in the revised manuscript are evidenced in red.

The only note I can make is to have inserted redundant information about the 'anatomical theater' and the 'figure of the assistant', which interrupt the discussion of the topic and do not make it easy for the reader to read.

According to the reviewer’s suggestion, these parts have been shortened.

Finally, it would be useful to include in the bibliography some references relating to the question today much debated of the methods of teaching medicine. This would give a scientific basis to the statements, which in my opinion are quite correct, relating to the issue of teaching/research/publications.

In the “Discussion and conclusions” section a new paragraph dealing with the reviewer’s request was included.

Reviewer 2 Report

Many bibliographical data are missing (line 26, 74, 77, 248  316 and foll.: place of printing and edition).

Line 224: Rota's pupils: the expression sounds strange

Line 233: Concordi: what is it? where is it?

Line 293: the name of the hospital should not be translated as they have done in the other cases. only one rule must be followed

Author Response

Referee 2

The authors would like to thank the reviewer for the precious suggestions to improve the manuscript. Changes in the revised manuscript are evidenced in red.

Many bibliographical data are missing (line 26, 74, 77, 248 316 and foll.: place of printing and edition).

Some books mentioned in the text have been fully reported in the reference list. For other references it was not possible to find full information. The work by Aldovrandi Ornithologiae hoc est de avibus historiae includes three volumes and each mentioned volume is cited separately in “references”.

Line 224: Rota's pupils: the expression sounds strange

According to the reviewer’s suggestion, “pupil” was replaced with “student”.

Line 233: Concordi: what is it? where is it?

The Accademia dei Concordi was founded in Rovigo (Venetia, Italy) in 1580, dealing with art and literature. It was one of the several “accademie” founded since Italian Renaissance to protect and disseminate culture. Some of them became more famous and important, as the “Accademia del Cimento” founded by Galilei’s pupils in 1657. The Accademia dei Concordi, still existing, is a minor cultural society. Some words have been introduced in the text of the manuscript to explain this accademia.

Line 293: the name of the hospital should not be translated as they have done in the other cases. only one rule must be followed

The name of the hospital was not translated.

Reviewer 3 Report

This is indeed an interesting manuscript which both describes a less known historical figure - quite important for the overall history of anatomy - and offers a reflection on the real role and impact of publications in determining a scientist's fate. The manuscript has some little flaws that, once edited, will make the paper look better:

  • the English is generally good but certain sentences do not flow naturally nor are certain choices standard (e.g. 'XV century' instead of '15th century'; 'galenism' instead of 'Galenism'). I strongly recommend having it read by a native speaker;
  • The author/s discuss too abruptly the standing and history of anatomical teaching at Bologna, without mentioning the earlier, Mediaeval phases: a cursory mention of Mondino de' Liuzzi's role and work would help readers unfamiliar with the context presented by the author/s. I would add the following two recent references, which thoroughly discuss the origins and development of the anatomical sciences: 1) Papa V, Varotto E, Vaccarezza M, Ballestriero R, Tafuri D, Galassi FM. The teaching of anatomy throughout the centuries: from Herophilus to plastination and beyond. Medicina Historica 2019;3(2):69-77 and 2) Papa V, Varotto E, Vaccarezza M, Galassi FM, 2021: Teaching anatomy through images: the power of anatomical drawings. Anthropologie (Brno) 59, 2: 145-153.
  • the whole discussion about 'publish or perish' is extremely important but, in my view, much too hurried in its presentation. It should also include at least a mention of the Latin==>French==>German==>English  historical transition in the languages used to communicate science. It should also try to briefly examine why Italian institutions publishing in Latin dominated the anatomo-surgical debate until the 16th century but gradually declined, while English-language institutions grew exponentially and published in English. Lines 55-58 in the introduction need at least one citation, otherwise they appear a personal statement. Some more lines should also be dedicated to the role of Schools and Maestri and the importance of teaching in the past. This should also taken into account when considering that young scholars may have been deterred from publishing independently should their professor set a low publishing standard, etc. 

These minor edits will improve this paper, which I have appreciated very much. 

Author Response

Referee 3

This is indeed an interesting manuscript which both describes a less known historical figure - quite important for the overall history of anatomy - and offers a reflection on the real role and impact of publications in determining a scientist's fate. The manuscript has some little flaws that, once edited, will make the paper look better:

The authors would like to thank the reviewer for the words of appreciation regarding the manuscript. Changes in the revised manuscript are evidenced in red.

the English is generally good but certain sentences do not flow naturally nor are certain choices standard (e.g. 'XV century' instead of '15th century'; 'galenism' instead of 'Galenism'). I strongly recommend having it read by a native speaker;

These expressions have been corrected, also in the title.

The author/s discuss too abruptly the standing and history of anatomical teaching at Bologna, without mentioning the earlier, Mediaeval phases: a cursory mention of Mondino de' Liuzzi's role and work would help readers unfamiliar with the context presented by the author/s. I would add the following two recent references, which thoroughly discuss the origins and development of the anatomical sciences: 1) Papa V, Varotto E, Vaccarezza M, Ballestriero R, Tafuri D, Galassi FM. The teaching of anatomy throughout the centuries: from Herophilus to plastination and beyond. Medicina Historica 2019;3(2):69-77 and 2) Papa V, Varotto E, Vaccarezza M, Galassi FM, 2021: Teaching anatomy through images: the power of anatomical drawings. Anthropologie (Brno) 59,2:145-153.

This part has been included in the second chapter with appropriate references.

the whole discussion about 'publish or perish' is extremely important but, in my view, much too hurried in its presentation. It should also include at least a mention of the Latin==>French==>German==>English historical transition in the languages used to communicate science. It should also try to briefly examine why Italian institutions publishing in Latin dominated the anatomo-surgical debate until the 16th century but gradually declined, while English-language institutions grew exponentially and published in English.

This part has been included in the “Introduction” section with appropriate references.

Lines 55-58 in the introduction need at least one citation, otherwise they appear a personal statement.

A citation has been included.

Some more lines should also be dedicated to the role of Schools and Maestri and the importance of teaching in the past. This should also taken into account when considering that young scholars may have been deterred from publishing independently should their professor set a low publishing standard, etc.

Some lines were added in the second chapter.

These minor edits will improve this paper, which I have appreciated very much.

Again, the authors would like to thank the reviewer for the words of appreciation regarding the manuscript.